# Optical widefield nuclear magnetic resonance microscopy

Karl D. Briegel [1,2], Nick R. von Grafenstein [1,2,4], Julia C. Draeger [1,2,4], Peter Blümler [3], Robin D. Allert [1,2] & Dominik B. Bucher [1,2] ✉

Microscopy enables detailed visualization and understanding of minute structures or processes. While cameras have significantly advanced optical, infrared, and electron microscopy, imaging nuclear magnetic resonance (NMR) signals on a camera has remained elusive. Here, we employ nitrogen-vacancy centers in diamond as a quantum sensor, which converts NMR signals into optical signals that are subsequently captured by a high-speed camera. Unlike traditional magnetic resonance imaging, our method records the NMR signal over a wide field of view in real space. We demonstrate that our optical widefield NMR microscopy can image NMR signals in microfluidic structures with a ~10 μm resolution across a ~235 × 150 μm² area. Crucially, each camera pixel records an NMR spectrum providing multicomponent information about the signal's amplitude, phase, local magnetic field strengths, and gradients. The fusion of optical microscopy and NMR techniques enables multifaceted imaging applications in the physical and life sciences.

Cameras have revolutionized microscopy, spanning visible light, fluorescence, infrared, and electron microscopy. In contrast to scanning methods, widefield detection with cameras increases speed and throughput by performing parallel measurements. However, the use of cameras in nuclear magnetic resonance/imaging (NMR/MRI) techniques has been hampered by their inability to detect magnetic resonance signals directly. In this study, we use nitrogen-vacancy (NV) centers in diamond as microsensors to convert local NMR signals into optical signals[1–9] that can be detected by a camera in real space, allowing parallel widefield imaging. This approach differs significantly from conventional MRI[10,11], which relies on encoding spatial information via magnetic field gradients. We use a high-speed camera to stroboscopically interrogate the NMR signal encoded in the fluorescence intensity of an NV center-doped diamond chip. This is in strong contrast to previous NV-based work, where either static magnetic fields[12,13] or nanoscale NMR signals[14,15] were imaged, and time-resolved measurements were not needed. Importantly, nanoscale liquid state NV-NMR experiments are typically limited by broad resonance lines which is not a limitation in our present work[16] (Supplementary Note 1).

We first establish a high-speed camera readout for a quantum sensing protocol designed for NMR signal detection. Subsequently, we apply this technique to image NMR signals from within the micro-structures of a microfluidic chip, achieving a spatial resolution of ~10 μm over a field of view of ~235 × 150 μm². Each pixel within the captured images provides information on the local amplitude and phase of the NMR signal, exhibiting a strong dependence on the local geometry, as validated by our simulations. In addition, the NMR signal provides information about local magnetic fields and gradients. Thus, our optical widefield nuclear magnetic resonance microscopy (OMRM) effectively bridges the gap between optical microscopy and information-rich magnetic resonance methods.

## Results

### Principle of operation and experimental design

Nuclear spins (e.g., ¹H nuclei) can be excited by a radiofrequency (RF) pulse, causing them to precess around an applied magnetic field $B_O$ and emit an RF/NMR signal at their Larmor frequency (Fig. 1a). Instead of detecting the NMR signal inductively, we place the sample on an NV center-doped diamond chip. By coherent microwave (MW) control of

[1]Technical University of Munich, TUM School of Natural Sciences, Department of Chemistry, Lichtenbergstraße 4, 85748 Garching bei München, Germany. [2]Munich Center for Quantum Science and Technology (MCQST), Schellingstr. 4, 80799 München, Germany. [3]University of Mainz, Institute of Physics, Staudingerweg 7, 55128 Mainz, Germany. [4]These authors contributed equally: Nick R. von Grafenstein, Julia C. Draeger. ✉e-mail: dominik.bucher@tum.de

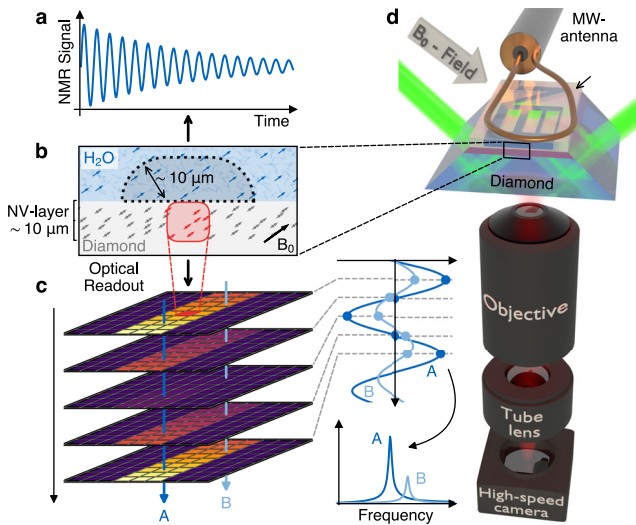

**Fig. 1 | Basic principle of optical widefield nuclear magnetic resonance microscopy. a** Nuclear magnetic resonance (NMR) signals are caused by the precession of nuclear spins (e.g., $^1H$) around an applied magnetic field $B_O$ after excitation by a radiofrequency pulse. **b** The NMR signal is detected optically using nitrogen-vacancy (NV) centers as local magnetometers in the diamond. We use a ~10 μm thick NV center-doped diamond layer, which sets the spatial resolution to a similar length scale. **c** The fluorescence intensity encoded NMR signal is imaged with a high-speed streaming camera. Each pixel time trace, corresponding to a location on the diamond, can be Fourier transformed (FFT) into a frequency spectrum. **d** Experimental setup: The NV layer of the diamond is optically excited at 532 nm using a total internal reflection geometry. The NMR sample is in a microfluidic structure on top of the diamond. For sensing, the NV spins are coupled to the sample spins by microwave pulses from an antenna. The resulting NV's spin state-dependent fluorescence intensity is captured by an objective and imaged with a tube lens onto a high-speed streaming camera. The probe head is positioned within a highly homogeneous magnetic field $B_O$ at ~84 mT for NMR detection.

the NV centers, we couple their electronic spin states to the NMR signal[16,17]. The radius of the dominant detection volume, which determines the spatial resolution, roughly corresponds to twice the average depth of the NV center ensemble from the diamond surface[1,18] (Fig. 1b). Since the NV center's fluorescence intensity is spin state dependent[17,19], it converts the NMR signal from a magnetic to an optical signal (Supplementary Note 2). By imaging the spatial fluorescence intensity of the NV diamond on a camera, we can record the transcoded NMR signal and thus obtain a spatially distributed map of local NMR spectra (Fig. 1c). Thus, the spectra are recorded in parallel for each camera pixel and do not require further encoding steps or magnetic field gradients to obtain spatial information.

The core of the OMRM microscope is a ~500 μm thick electronic grade diamond with a ~10 μm overgrown NV center-doped diamond layer, which defines the spatial resolution (Fig. 1d, Supplementary Note 3). The diamond is polished into a trapezoidal shape in order to couple in the green excitation light (532 nm) in a total internal reflection geometry. We use a laser spot to illuminate the NV layer over an area of ~235 × 150 μm². The diamond is bonded into a custom-designed microfluidic glass chip[20]. To control the NV spinstate, MW pulses are delivered via a short-circuited coaxial cable placed on top of the microfluidic chip[21]. A 3D-printed sample holder houses the microfluidic chip and mounts a pair of copper wire coils for the RF excitation of the proton sample spins. The entire probe head, made of non-magnetic materials, is mounted in a custom-built highly homogeneous permanent magnet[22]. A high numerical aperture objective with low magnetic properties and tube lens are used to collect and image the fluorescence of the NV layer onto a high-speed streaming camera. Residual excitation light is filtered with a long pass filter to retain the signal-carrying fluorescence (details in Methods). With this setup, we

achieve a sensitivity of ~10–30 nT Hz$^{-1/2}$ μm$^{3/2}$ for RF signals (Supplementary Note 4).

## NV quantum sensing over a wide field of view
In all measurements, the NV center's electron spin state is optically initialized by laser excitation. MW pulse sequences are subsequently used to manipulate the NV spin for sensing over the field of view, and the change in the spin state of the NV is optically imaged[17] (Fig. 2a). In the first experiment, we determine the NV Rabi frequency, which forms the basis of the pulse sequence for detecting NMR signals. By applying progressively longer MW pulses at the NV resonance frequency, we rotate the NV spin state and extract the per-pixel spin control parameters that lead to half and full spin flips, i.e., π/2- and π-pulses (Fig. 2b). Color coding of the π-pulse duration provides a map of the MW driving field strength (Fig. 2c), which varies slightly across the field of view, due to the geometry of the MW antenna. We observe an additional pattern that may originate from local laser-induced temperature variations. However, as it does not seem to impact the following sensing schemes (Fig. 2f), the origin was not investigated further.

To detect high frequency-resolved NMR signals with NV centers, we use the coherently averaged synchronized readout (CASR) pulse sequence[1,23,24] (Supplementary Note 2). CASR is based on a precisely timed train of π-pulses that couples the NV center's spin to the NMR signal by matching the repetition rate of the π-pulses to half the period of the expected NMR signal frequency (dynamical decoupling sequence[17,25], Fig. 2d). For a given NMR signal and MW pulse train, the magnitude of the coupling is a function of the phase of the NMR signal relative to the π-pulse train. A precisely timed repetition of the coupling and readout process results in a time-varying coupling amplitude at an aliased frequency of the original NMR signal, which can be detected optically using the NV photoluminescence (Figs. 1b and 2e). In previously published NV-based magnetic imaging studies, the experimental procedure allows repetition and averaging over individual sweep parameters, such as the MW frequency in quantum diamond microscopy or the pulse duration in Rabi experiments[12–14,17,26–29]. However, when the CASR pulse sequence is used for NMR signal detection, real-time readout (~5000–20,000 frames/s) becomes essential[30,31] (Supplementary Note 5). This is due to the inherent synchronization requirements between the CASR pulse sequence and the (decaying) NMR signal (Figs. 1c and 2d). This is trivial for a photodiode but poses technical challenges for imaging systems. Our solution is a high-speed streaming camera with a framerate of up to multiple tens of kHz (Methods, Supplementary Note 5). We use a frame rate of 6000 fps for all measurements in the main text.

Secondly, we demonstrate the detection of continuous RF calibration signals from a nearby antenna emitting at ~3.56 MHz. Each pixel records a photoluminescence (PL) oscillation at the aliased RF frequency in the time domain (Fig. 2e, inset), which is Fourier transformed. Color coding of the corresponding signal-to-noise ratio (SNR) yields an image (Fig. 2f, Supplementary Note 6). We observe a similar large scale spatial variation of the SNR as in the Rabi image. The two are related because well-defined π-pulse durations are required for the CASR pulse sequence, and variations will reduce its sensitivity. For this reason, the field of view is currently limited by the homogeneity of the Rabi frequency distribution (MW field homogeneity, Fig. 2c, f), which can be mitigated in the future by using MW resonators[16].

## Optical widefield nuclear magnetic resonance microscopy (OMRM)
In the next step, we utilize the CASR protocol to detect NMR signals from a water sample contained within the microfluidic structure. We apply Overhauser dynamic nuclear polarization (ODNP) to increase the NMR signal by two orders of magnitude at our low magnetic fields (~84 mT, $^1H$ NMR at ~3.56 MHz)[32,33]. An RF pulse induces the NMR signal

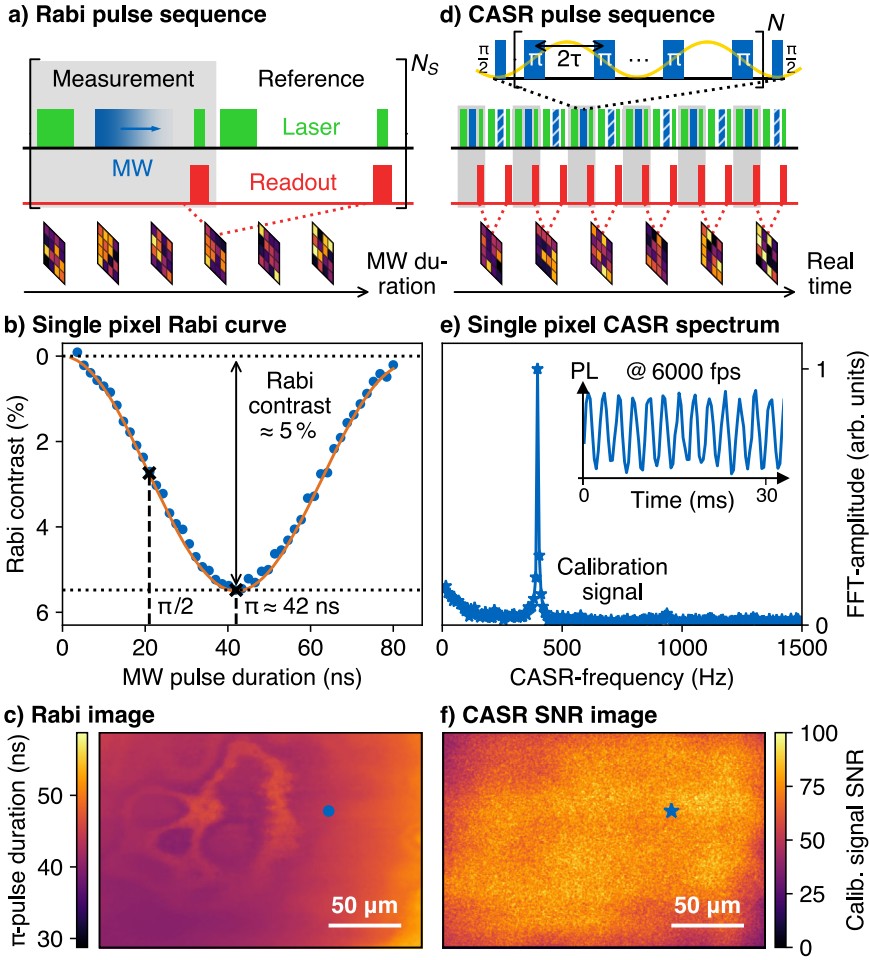

**Fig. 2 | Widefield quantum sensing with a high-speed streaming camera.** The employed nitrogen vacancy (NV) center quantum sensing protocols require an initialization laser pulse (green), a microwave (MW) pulse (blue) to control the NV spin state, and a second laser pulse to read out the spin-dependent photoluminescence (PL) of the NV (red), which is captured by a camera. **a** For a Rabi experiment, the NV center's spin state is observed as a function of the MW pulse duration. For each MW pulse duration, the spin state-dependent PL, as well as one reference image for common mode noise rejection, are recorded and averaged $N_s$ times. **b** Increasing the MW pulse duration leads to spatially resolved Rabi oscillations for each pixel. The π/2- and π-pulse durations, determined for each pixel, serve as important control parameters for the coherently averaged synchronized readout (CASR) sequence to detect nuclear magnetic resonance (NMR) signals. **c** Image of the π-pulse duration determined from the single pixel Rabi fits. The pixel corresponding to the dataset in (**b**) is marked with a blue dot. **d** Radiofrequency (RF) signals are detected stroboscopically using the CASR pulse sequence. CASR is based on blocks of MW pulses (blue) starting and ending with an π/2- and $N$ trains of π-pulse coupling the NV centers to the detected RF field (yellow). After each sensing block, the NV-fluorescence is detected, requiring a time-resolved readout on a high-speed streaming camera. **e** Detection of an RF calibration signal. The recorded pixel-wise time domain data (inset: a snapshot of the time domain signal) is Fourier transformed and shows a signal at the aliased CASR frequency. **f** Image of the signal-to-noise ratio (SNR) of the single pixel analysis of the CASR data. The pixel corresponding to the dataset in (**e**) is marked with a blue star.

which is detected using the CASR pulse sequence (Fig. 3a). In contrast to the RF calibration signal, the NMR signal has a limited lifetime ($T_2$/$T_2^*$) after the excitation. To maximize the lifetime of the NMR signal, we use a highly homogeneous magnetic field $B_O$ from a custom-designed permanent magnet and purposefully chosen non-magnetic components for the probe head (details in Methods).

In our first OMRM example, we image the NMR signals of water within the marked area of the microfluidic chip (Fig. 3b) with a total measurement time of ~14 to ~16 h (including all dead times). Applying the Fourier transformation (FFT) along each pixel's time domain followed by a 2D median filter over the resulting frequency images yields single pixel NMR spectra (Fig. 3c–e) (Supplementary Note 7). Color coding the SNR of the NMR signal results in an OMRM image that clearly depicts the microfluidic channel filled with water (Fig. 3f, Methods). Each pixel on the camera - encoding the spatial information - contains a full NMR spectrum. The spatial resolution of ~10 μm is currently dominated by the magnetic resolution, which is a function of the NV layer thickness (~10 μm) (Fig. 3g). This is due to the dipolar

interaction of the NV and the sample spins[18], which is in agreement with our simulations and additional experiments performed with a thicker NV layer (Supplementary Note 3). Optical aberrations imposed by imaging through our diamond substrate likely play a minor role[34].

**Multicomponent analysis of the OMRM image**
For more detailed analysis, we fit a Lorentzian function (Eq. (1)) to each pixel's complex valued NMR spectrum resulting from the FFT of the time domain signal (full data analysis and complementary analysis can be found in Supplementary Note 7). This yields information about the signal amplitude $S_O$, phase $\phi$, frequency $f_O$, and linewidth $\Gamma$ of the NMR signal for each pixel.

$$S(f) = S_0 \cdot e^{i\phi} \cdot \frac{\Gamma - i \cdot (f_0 - f)}{\Gamma^2 + (f_0 - f)^2} \qquad (1)$$

To analyze parameters other than signal size more accurately, we remove unreliable fits if their raw data SNR is below a certain threshold (Supplementary Note 7). Using the calibration signal amplitude as a

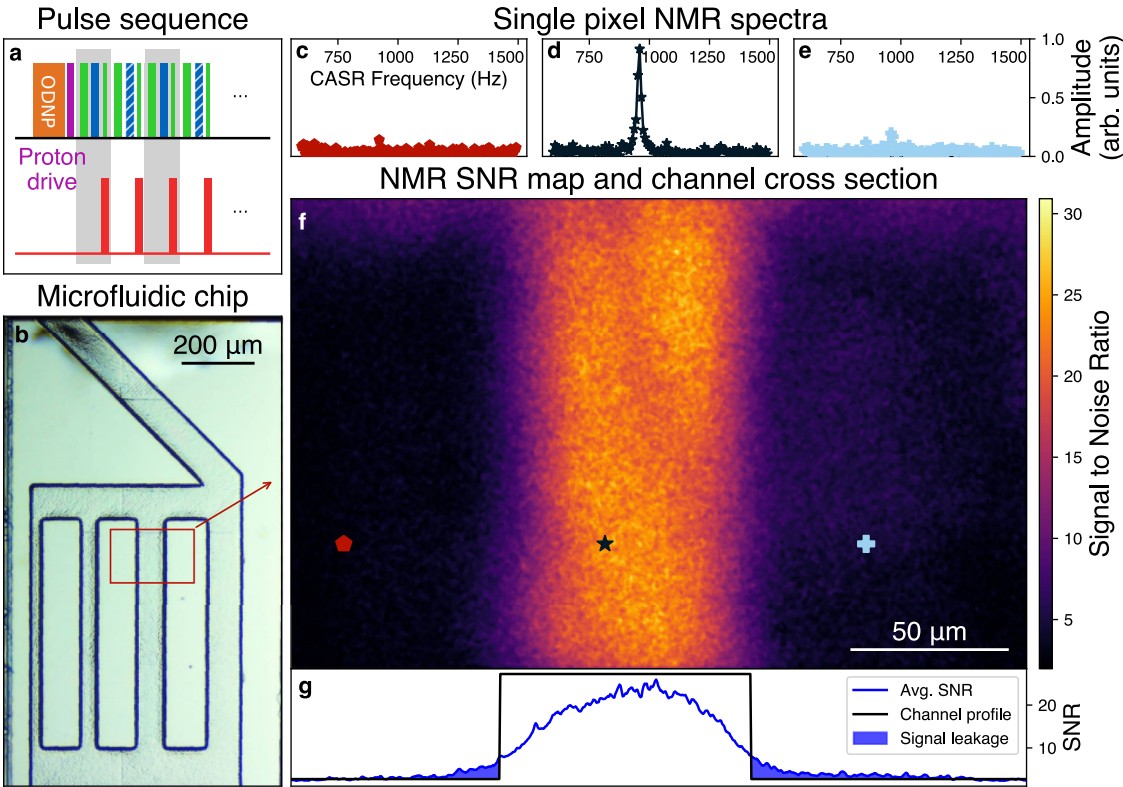

**Fig. 3 | Optical widefield nuclear magnetic resonance microscopy (OMRM). a** NV-NMR pulse sequence. The (orange) Overhauser dynamic nuclear polarization (ODNP) and (magenta) [1]H excitation pulses are followed by the coherently averaged synchronized readout (CASR) pulse sequence, which converts the nuclear magnetic resonance (NMR) signal into an optical signal. **b** Optical microscopy image of the microfluidic chip. Our water sample is pumped and distributed through microfluidic channels through an inlet on the top part of the chip.

**c–e** Single-pixel NMR spectra at the marked locations in (**f**) vs CASR-frequency ([1]H NMR frequency at ~3.56 MHz aliased to ~960 Hz). **f** Signal-to-noise ratio (SNR) map of the NMR spectrum obtained by Fourier transforming the single pixel NMR signals. The SNR encodes the presence of the [1]H sample in the microfluidic chip. The data was recorded from the red region in (**b**). **g** Channel cross-section of the average SNR, indicating the spatial resolution for our ~10 μm NV-layer.

reference for the [1]H NMR signal amplitude $S_O$ mitigates laser and MW-drive inhomogeneities and allows us to reconstruct the channel path of the microfluidic chip (Fig. 4a, b). Due to the dipolar nature of the spin interaction and its associated symmetries, the amplitude of the NMR signal is dependent on the geometry of the NMR sample relative to the NV center[18]. In Fig. 4a, the axis of the NV center and the magnetic field $B_O$ are orthogonal to the vertical microfluidic channel (left). This leads to an asymmetric signal amplitude across the channel width, which is confirmed by our simulations (Fig. 4b, bottom, Supplementary Note 8). Shifting the field of view to a different region of the microfluidic chip results in a channel that runs at an angle of ~45° relative to the NV center's axis (Fig. 4a, middle). This channel orientation changes the interaction symmetry of the NV centers, eliminating the signal amplitude asymmetry in agreement with our simulations. A final position on the microfluidic chip with a T-junction of the channels further supports this observation (Fig. 4a, right). The channel running orthogonal to the NV center's principal axis reproduces the observed asymmetry in $S_O$.

An additional parameter is the phase $\phi$ of the NMR signal, which also depends on the geometry and varies accordingly across the microfluidic channels (Fig. 4c). While the origin of this observation is the same as before, the observed gradients in $\phi$ follow the opposite trend compared to $S_O$. Figure 4c shows the same channel sections as in Fig. 4b; however, the signal phase is symmetric around the vertical channel (Fig. 4c, left). Shifting the field of view to the diagonal section of the channel changes the interaction symmetry and introduces a phase gradient across the channel (Fig. 4c, middle). The T-junction reproduces the symmetric phase distribution around the orthogonal

portion of the channel, while the parallel part exhibits a phase gradient across its width (Fig. 4c, right).

The signal frequency $f_O$ provides information about the Larmor frequency of the sample, which can give important chemical information. Since our sample is water, which exhibits a single [1]H resonance, $f_O$ serves as a highly accurate measure of the local magnetic field strength across the field of view (Fig. 4d). In our measurements, the observed gradient was ~6 nT/μm for all chip positions, indicating a macroscopic magnetic field gradient.

Finally, we can resolve the local NMR linewidth $\Gamma$, which provides information about the proton relaxation time. We observe systematic changes in $\Gamma$ across the field of view, which could be caused by local magnetic field gradients induced by susceptibility mismatches.

We note that the data had to be corrected for slow magnetic field $B_O$ drifts caused by the temperature-dependent magnetization of the permanent magnet. To correct for this frequency drift, we store the acquired data every ~5 min and frequency correct the time domain NMR signal (Methods, Supplementary Note 7). Therefore, the amplitude and linewidth values obtained should not be compared between different measurements (Supplementary Note 7).

## Discussion
These measurements constitute a demonstration of microscale optical widefield NMR microscopy with rich information content. Presently, our spatial resolution is limited by the NV layer thickness of ~10 μm (Supplementary Note 3). The spatial resolution of the OMRM technique is technically limited by the optical diffraction limit and the NMR sensitivity, where the latter scales proportionally with the number of

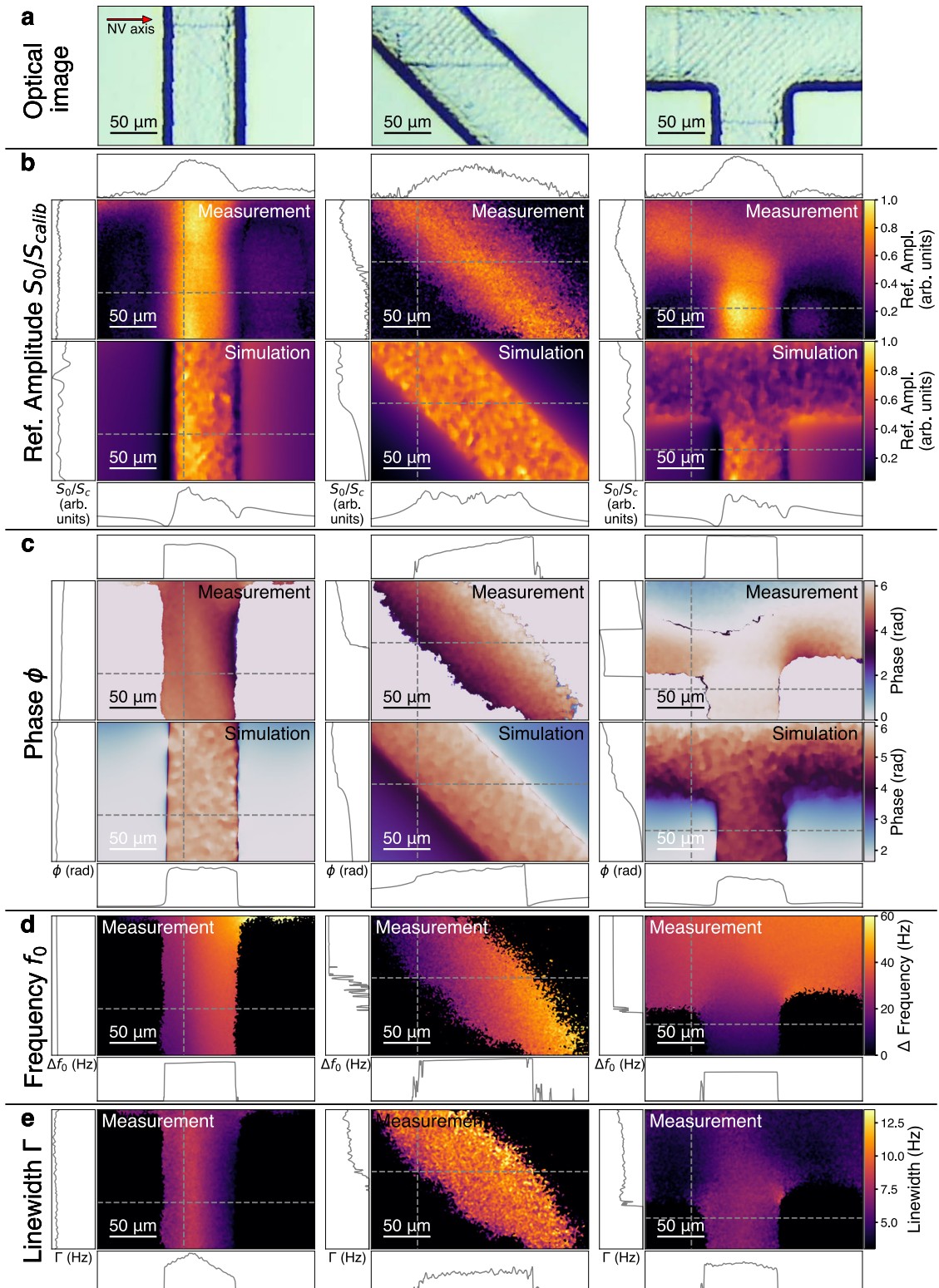

**Fig. 4 | Multicomponent optical widefield NMR microscopy. a** Optical photographs depicting microfluidic chip sections that were subsequently imaged with OMRM in (**b**–**e**) (left, middle, right). **b** Top row: Referenced nuclear magnetic resonance (NMR) signal amplitude ($S_0/S_{Calib}$, Supplementary Note 7) image obtained from a Lorentzian fit at three different locations on the microfluidic chip (**a**). Dotted lines indicate a cut through the image, plotted adjacent to them. Bottom row: Corresponding simulated amplitude maps for the same channel regions

indicate an influence of the geometry on the NMR amplitude. **c** Top row: Fitted relative phase $\phi$ image of the NMR signal. Bottom row: Corresponding simulated phases of the NMR signals indicate an influence of the geometry on the NMR phase. **d** Image of the NMR signal frequencies $f_O$ across the field of view indicate a macroscopic gradient in $B_O$. **e** Linewidth $\Gamma$ of the NMR signal indicates local magnetic field gradients. The full data processing can be found in Supplementary Note 7.

NV centers per pixel (Supplementary Note 9). For the current setup and spatial resolution, we obtain a sensitivity of ~10–30 nT Hz$^{-1/2}$ μm$^{3/2}$ with vast optimization possibilities in the future (detailed discussion in Supplementary Note 10). With these improvements, achieving single-micrometer spatial resolution will be feasible.

In contrast to previous nanoscale NV-NMR imaging work[14,15], our spectral resolution of ~ 3 ppm is only limited by the sample's coherence time[1]. We are currently limited by magnetic field homogeneity and stability, which can be greatly improved through further engineering (e.g., by using a superconducting magnet[35]). Thus, we envision chemical imaging on a per-pixel basis for highly parallelized NMR analysis in the future.

In summary, our results demonstrate the fusion of optical wide-field microscopy with NMR spectroscopy, synergistically combining the strengths of each technique: the high-speed and high spatial resolution of optical imaging on a camera, together with the unique and comprehensive chemical and physical insights provided by magnetic resonance methods. In the future, we expect to increase the wealth of information by incorporating ($T_1$ and $T_2$) relaxation, diffusion[35], and (multinuclear) high spectral resolution spectroscopy for structural molecular information[1,6]. As a result, our methodology will provide unprecedented insights into the microscopic world, with diverse applications ranging from (metabolomic) analysis of single cells[36] or tissues to high-throughput NMR spectroscopy and materials science[37].

## Methods

### NV diamond ensemble sensors

The NV diamond NMR sensors were made from single crystal chemical vapor deposition (CVD) diamond substrates with a bulk nitrogen concentration of <5 ppb and a {100} face orientation (Element Six Technologies Limited, Didcot, United Kingdom). The first sample was overgrown with a nitrogen-enriched layer (~10 μm with ~25 ppm $^{14}$N and ~99.999% $^{12}$C) by Element Six Technologies Limited (Didcot, United Kingdom). NV centers were generated by electron irradiation (~2.3·10$^{18}$ e$^-$/cm$^2$) at 4.6 MeV, followed by low-pressure-high-temperature (LPHT) annealing (<10$^{-7}$ mbar, 800 °C for ~12 h). The second NV-diamond sample was CVD-overgrown with a nitrogen-doped-layer (~40 μm with ~19 ppm $^{15}$N and ~99.99% $^{12}$C) at the Fraunhofer Institute for Applied Solid State Physics (IAF, Freiburg, Germany)[38]. The substrate was electron irradiated with ~1·10$^{18}$ e$^-$/cm$^2$ at ~1 MeV (US Diamond Technologies, New York, United States) and annealed (<10$^{-7}$ mbar, 800 °C for 16 h). Both NV-diamond samples were polished at a 45° angle on two sides, yielding a trapezoidal-shaped diamond where the face containing the NV centers has a size of ~2.0 mm × 1.0 mm.

### Permanent magnet

The permanent magnet (Supplementary Fig. 16) used to produce the ~84 mT bias field has the layout of a Halbach dipole[39,40]. It was built analogously to the procedures described by Wickenbrock et al.[22] with differences in size and material to reach the target magnetic field and to fit the OMRM microscope inside. We used a special Sm$_2$Co$_{17}$ alloy provided by EEC (Electron Energy Corporation, Landisville, PA, USA) with an extremely low-temperature dependence of ~0.001%/°C of its remanence (Specifications: Material Grade: EEC 2:17-TC16, typical remanence ~84 mT, min: ~78 mT; intrinsic coercivity ~1910 kA/m) to provide a stable magnetic field despite temperature fluctuations in the laboratory. Using these parameters, the optimal design was determined analytically[40] and by FEM-simulations (COMSOL Multiphysics 5.4) to result in the following geometry: Two rings consisting of 16 cylindrical permanent magnets, each aligned axially with an optimized gap between them. The 32 cylindrical magnets had a diameter of ~24 mm and a height of ~32 mm. Their centers were mounted on a circle with a radius of ~70.02 mm. The gap between the two rings was calculated to be ~28.3 mm (magnet-to-magnet surface) and later optimized manually for the highest homogeneity. The 32

magnets were chosen from a total order of 45. After their individual strength was determined[22], their positions in the final magnet arrangement were optimized by a Monte-Carlo algorithm[40]. They were then glued in cylindrical segments (Supplementary Fig. 16b) while being magnetically oriented as described in Wickenbrock et al.[22] After manual fine-tuning, the magnet provided a magnetic field $B_O$ of ~84 mT. The full magnet assembly is shown in Supplementary Fig. 16d, e. The magnet is mounted to a post with a manual rotation stage (RP01/M, Thorlabs, Newton (New Jersey), USA) to enable tilting of the magnet to align one NV center's principal axis in the diamond lattice with the magnetic field. The post is placed on an XY stage (2x XR25P/M, Thorlabs, Newton (New Jersey), USA) and a heavy-duty lab jack (L490/M, Thorlabs, Newton (New Jersey), USA) for control of its position.

### Magnetic field homogeneity

The permanent magnet provides the highly homogeneous magnetic field necessary to detect NMR signals. All parts in the vicinity of the magnet (optics, cables, screws, etc.) are selected to be non-magnetic. Procurement of non-magnetic objectives is particularly cumbersome, as most vendors provide little information about the magnetic properties of their objectives. For this reason, we tested different suppliers and chose the MXPlanFLN (50×, 3 mm WD, Olympus, Shinjuku, Japan) because it had the least influence on the magnetic field homogeneity and, consequently, on the NMR line width. Nevertheless, we observe significant line broadening, which is likely caused by residual magnetization of the objective.

### Microfluidic structure/chip

The microfluidic chip was designed with computer-aided design (CAD) software (FreeCAD) and manufactured by LightFab GmbH (Aachen, Germany)[20]. The chip's central structure, adjacent to the diamond, contains one diagonal channel (~100 μm width) and four parallel channels with ~100, ~80, ~40, and ~20 μm width and ~80 μm height (Supplementary Fig. 17a).

### Sample holder

The trapezoidal NV-diamond and the microfluidic tubing (SGEA13010050015F, VWR, Randor, United States) were glued into the microfluidic chip using optical adhesive (NOA68, Norland Products, Jamesburg, United States), which was cured for >12 h at ~60 °C under UV illumination (Supplementary Fig. 17a, b and supplementary Fig. 18a). The inlet is connected to a 1 mL syringe containing the NMR sample, which is mounted into a syringe pump (Al-1000, World Precision Instruments, Sarasota (Florida), USA) set to a flow speed of ~3–15 μl/h to replenish the hyperpolarizing agent bleached by the laser (TEMPOL, 581500, Sigma-Aldrich, St. Louis, USA). The sample is guided through a bubble trap (LVF-KBT-S, Darwin Microfluidics, Paris, France) to prevent air bubbles from getting trapped in the microfluidic chip. The diamond-microfluidic assembly is placed in a custom-designed 3D-printed sample holder (Formlabs, Massachusetts, USA) Form-3, Formlabs gray resin). To provide the calibration RF signal, a coil with a diameter of ~10 mm is placed in the diamond vicinity. To drive the sample nuclear spins with a homogeneous field, two parallel coils with a diameter of ~18 mm and 6 turns per side are added (Supplementary Fig. 17c and Supplementary Fig. 18b, c).

### Optical setup

A 532 nm green laser (Verdi G7, Coherent, Saxonburg (Pennsylvania) USA) was passed through a λ-half-wave plate (WPH10M-532, Thorlabs, Newton (New Jersey), USA) to optimize the polarization for the acousto-optic modulator's (AOM) efficiency. A set of two 400 mm lenses (LA1172-A-ML, Thorlabs, Newton (New Jersey), USA) are used to focus the laser into the AOM (AOM - 3250-220 S/N, AOM driver 1250AFP-D-6.6, Gooch & Housego, Ilminster, UK) and to collimate it

afterward. The AOM (and an aperture blocking all but the first-order diffracted laser beam) allows for precise laser pulsing timed by TTL signals from the arbitrary waveform generator (AWG). The laser power after AOM was ~3 W. A second λ-half-wave plate is used to adjust the laser polarization for maximal Rabi contrast and a 100 mm lens (LA1509-B-ML, Thorlabs, Newton (New Jersey), USA) focuses the laser beam on the trapezoidal NV-diamond chip in a total internal reflection geometry. By adjusting the lens position, the size of the excitation area is controlled. The sample holder housing the microfluidic chip and the diamond is mounted in a 30 mm optical cage system (Thorlabs, Newton (New Jersey), USA) hanging from a frame made of 66 mm optical rails (XT66 66 mm Construction Rail, Thorlabs, Newton (New Jersey), USA). This allows the sample holder to be freely rotated and adjusted in height. An objective (50×, 3 mm working distance, MXPlanFLN, Olympus, Shinjuku, Japan) is mounted below the diamond to collect the emitted fluorescence. A 100 mm tube lens (TTL100-A, Thorlabs, Newton (New Jersey), USA) was chosen to adjust the magnification by a factor of ~0.55 at the expense of slight image distortion. A long pass filter (LP01-647R-25, Semrock) blocks the excitation wavelength from reaching the high-speed streaming camera. The objective, tube lens, and long pass filter are mounted to the camera. The sample holder is not connected to the imaging optics or camera, allowing the imaging system to move freely relative to the sample. An LED (654/50 mm, HexaCube) positioned above the diamond is used to illuminate the microfluidic structure for optical imaging (Supplementary Fig. 19). To verify and adjust the imaging area and focus, the LED is turned on, the long pass wave filter removed, and the camera-objective assembly moved accordingly. After the long pass filter is added back in, the laser is turned on and positioned on the area to be imaged (Supplementary Figs. 19, 20).

### Camera

For imaging, a high-speed streaming camera is used (EoSense1.1MCX12-CM, SVS-Vistek GmbH, Gilching, Germany). Technical details can be found in Supplementary Note 5. The camera and the imaging optics were mounted onto an xyz-stage, to enable precise positioning with respect to the diamond sensor. The imaging optics were mounted directly onto the camera and positioned relative to the sample.

### MW and RF electronics

Precise spin control of the NV centers for quantum sensing requires well-defined microwave (MW) pulses. Pulse sequences are synthesized digitally at low frequency (500 MHz) using an arbitrary waveform generator (AWG) (AWG5202, Tektronix, Beaverton (Oregon), USA) and subsequently up-converted with an IQ (IQ-Mixer, MMIQ0218LXPC 2040, Marki Microwave, Morgan Hill (California), USA) and a local oscillator (SMB100A, Rohde & Schwarz, Munich, Germany). A broadband 50 W amplifier (AMP1016, Exodus, Las Vegas, USA) amplifies the MW signal delivered through a home-build MW-antenna[21] (using a UT-047C-TP (Micro-Coax, Pottstown (Pennsylvania), USA), positioned on top of the microfluidic channel. The MW electronics are designed for simultaneous driving of the NV centers (~5.2 GHz) and the electronic spin of the hyperpolarization agent (~2.3 GHz)[32]. Radiofrequency pulses for driving the sample's proton spins are generated with a signal generator (AFG1062, Tektronix, Beaverton (Oregon), USA) and amplified using a 30 W amplifier (LZY-22+, Mini-Circuits, Brooklyn (New York), USA). Using a pair of coils (~18 mm diameter with 6 turns each) wound around holders on the 3D-printed sample holder, we achieve ${}^1$H Rabi frequencies of ~1.3 kHz. The RF calibration signal was generated by a signal generator (DG1022z, Rigol Technologies, Suzhou, China) and supplied through an additional coil with a diameter of ~10 mm on the sample holder. The entire experiment (RF, MW, AOM pulses, and camera readout) was synchronized by the AWG (Supplementary Fig. 21).

### NMR sample

The ${}^1$H NMR signal from water (Milli-Q® Direct 8 water purification) was used. To increase the NMR signal, 4-Hydroxy-2,2,6,6-tetra-methylpiperidinyloxyl (TEMPOL, ~20 mM, 581500, Sigma-Aldrich, St. Louis, USA) is added as a hyperpolarizing agent for Overhauser dynamic nuclear polarization[32].

### Quantum sensing measurement parameters

Measurement parameters are described in Supplementary Note 12.

### Software

We use custom software as described in Supplementary Note 13.

## Data availability

The source data used in this study are available in the ZENODO database under accession code https://doi.org/10.5281/zenodo.13617350.

## Code availability

**QuPyt**, our measurement software, is available on GitHub at https://github.com/KarDB/QuPyt. **QuFit-rs**, which contains parts of our analysis software, is available on GitHub at https://github.com/KarDB/QuFit-rs. Our custom code, used for our simulations can be found on GitHub at https://github.com/KarDB/Micro_NV_NMR.

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

## Acknowledgements

This project has been funded by the Bayerisches Staatsministerium für Wissenschaft und Kunst through project IQSense via the Munich Quantum Valley (MQV), by the Federal Ministry of Education and Research (BMBF) as part of the VIP+ validation funding program (03VP10350) and the European Research Council (ERC) under the European Union's Horizon 2020 research and innovation programme (Grant Agreement No. 948049). The authors acknowledge further support by the DFG under Germany's Excellence Strategy–EXC 2089/1-390776260 and the EXC-2111 390814868.

## Author contributions

D.B.B. conceived and supervised the study. K.D.B. wrote the code for hardware control, pulse sequence generation, measurement execution, and data processing. K.D.B., J.C.D., and N.R.v.G. designed and built the experiment. K.D.B., J.C.D. conducted the experiments. R.D.A. contributed to various important parts of the experimental setup. P.B. and N.R.v.G. designed, built, and characterized the permanent magnet. K.D.B. performed the simulations. K.D.B., J.C.D., and D.B.B. analyzed and discussed the data and wrote the manuscript with inputs from all authors.

## Funding

## Competing interests

The authors declare no competing interests.
