## [Transparent Peer Review file · Nature Communications]

Optical Widefield Nuclear Magnetic Resonance Microscopy

Corresponding Author: Professor Dominik Bucher

Version 0:

Reviewer comments:

Reviewer #1

(Remarks to the Author)

The paper demonstrates imaging of NMR signals in microfluidic structures with a 10 μm spatial resolution, using widefield optical readout of a layer of NV centres in a diamond chip. Imaging of NMR signals using NV-diamond was previously reported but these previous demonstrations (Refs 9,26, 27) were limited in spectral resolution. Here the authors implement a protocol providing a dramatically improved resolution that in principle enables chemical identification. As such, it can be argued this paper reports the first genuine NMR imaging demonstration with NV-diamond. The amount of engineering that went into making these experiments possible is truly impressive, and extremely well documented in the supplementary material, which will make the paper a very useful contribution to the NV community. Overall, I think the paper a significant step in NV sensing technology. I recommend publication in Nat Comms after the authors have addressed my comments below. Many of them relate to details that are given in the supplementary material but in my opinion should be discussed in the main text.

1. In the introduction there should be an explicit acknowledgement of the previous NMR imaging demonstrations (Refs 9,26, 27) and a discussion of the spectral resolution and the ability (or inability) to detect chemical shifts in these previous works, and how the present work solves this limitation. This will help the reader to better appreciate the progress reported.
2. I think the conclusion is missing a quantitative discussion of the achieved spectral resolution, what it can enable beyond the water case (illustrative examples of chemical shifts that could be detected would be helpful), and what further improvements may be expected in the future.
3. Fig. 2c: comment on the spatial features observed in the left part, which are not expected from the loop antenna used. Are these measurement artefacts?
4. Fig 2e: add a time scale for the inset graph, and specify the frame rate and time resolution, since the high frame rate is a key technical difference compared to previous works.
5. Fig 2f: the SNR is plotted, and so the reader can't tell whether the spatial variations in SNR (by a factor of up to 2 it seems) are due to variations in signal or noise or both. I think it is important to show (possibly in the supplement) or at least comment on the flatness of the measured signal. Indeed, for an external calibration signal one would expect a uniform signal across the field of view, so variations in the measured signal would inform on the accuracy of the measurement. Probably related to this point is the fact the authors later (Fig. 4) normalise the NMR amplitude by the calibration signal amplitude. The reason for that normalisation should be explained.
6. Fig. 3d-f: Comment on why there is a residual NMR signal (small peak at the expected frequency). Is it due to bleeding of the signal from the channel due to the point-spread function?
7. The spatial resolution ($\sim 10 \mu\text{m}$) is a key achievement of the paper but there is no data in the main text to support the claim. I think the linecut from Extended Data Fig. 8 (the 10 μm case only) should be shown in Fig. 3 of the main text.
8. The total acquisition time for the NMR images shown in Figs. 2-4 should be given in the main text.
9. Fig. 4: the horizontal linecuts for the simulations are flipped in sign; this is very confusing and makes it hard to compare with experiment. Also the choice of some of the vertical linecuts is odd, some are mainly through the excluded data.

10. Fig. 4: What is the origin of the noise in the simulations?

11. The fitting procedure (Eq. 1) as explained in the main text is a bit confusing. The authors write “we fit a Lorentzian function (Eq. 1) to each pixel NMR spectrum” but Eq. 1 is complex-valued and not the usual real-valued Lorentzian and so it is not immediately clear how the parameters are extracted. If I understand correctly, Eq. 1 is fitted to the complex-valued FFT of the real-valued time-domain data. Maybe this could be clarified in the main text.

12. Fig. 4d: there is a gradient from left to right in the f_0 maps explained by non-uniformity of the applied B_0 field, but in the T-junction there is a less trivial variation between the vertical channel and the top reservoir. Please comment on the origin. If it is due to magnetic susceptibility variations as mentioned in the following paragraph on linewidth, I think the explanation should be moved to the discussion of the f_0 maps.

Reviewer #2

(Remarks to the Author)

In their manuscript, Briegel et al. introduce diamond-based wide-field nuclear magnetic resonance microscopy that makes use of a high-speed camera. The manuscript is well written and polished.

My main concern with this manuscript is that while the data is nice, it reads more like a methods paper, and the main advance is the use of a camera that allows fast data extraction. The novelty is not very high, as many aspects have been demonstrated already. Wide-field NMR using diamond has been performed before (e.g. see Ziem et al, Quantitative nanoscale MRI with a wide field of view, Scientific Reports volume 9 (2019)), and the group has already published on microscale NMR in microfluidics devices (Allert et al., Lab on a Chip 2022). Introducing a fast camera is interesting, and will be of interest to specialists in the community, but many of the presented results have been demonstrated before.

A few detailed comments, though overall the manuscript is very clearly laid out:

- I could not find any details on the two steps that precede the CASR pulse sequence (shown in Figure 3.a): Overhauser hyperpolarization and ^1H excitation pulses. I may have missed it but could not see anything in the supplementary materials. Please add details and calibrations for these.
- The discussion around the linewidth could be misleading, as the first sentence states this ‘provides information about the proton relaxation time’. In the regime reported here the linewidth is dominated by other effects, and it would be good to amend the statement, and perhaps provide estimates on what would need to be achieved for the linewidth to actually reveal proton relaxation times.
- In Figure 4.b) in the T geometry the simulation differs from measurement, particularly on the right hand side. Why is this?

Reviewer #3

(Remarks to the Author)

The authors demonstrate widefield optically detected nuclear magnetic resonance microscopy by using ensembles of NV centers in diamond read out by a high-speed streaming camera. They modify an established NMR sensing technique to be compatible with camera read out, and then use the platform to image proton NMR signals associated with water in microfluidic chambers with approximately 10 μm spatial resolution. They then match the measured NMR signal to a dipolar simulation and see reasonable agreement.

Previous work in the field has measured NMR signals from small volumes, interrogating either single NV centers or NV ensembles without spatial information. This technical work extends to a widefield camera readout with spatial information engineered with a microfluidic structure, which may be applicable for applications in condensed matter or biological physics. The manuscript is clear and well-written, and all of the claims are well-supported by the data. The results are a sensible step forward for widefield NMR sensing, and the authors extensively benchmark future improvements to sensitivity and spatial resolution, which will be useful to other researchers in the NV NMR field. I believe the paper merits publication as long as the authors address the following minor comments:

The authors claim that the NMR spatial resolution is limited solely by the NV layer depth, but aberrations from imaging through the diamond may also be a strong contribution. How thick are the diamonds used, and how much do aberrations from imaging through the diamond with the 0.8 NA objective affect the spatial resolution of the measurements?

The T_2^* of the detected protons was 30 ms, but the data in Fig. 4e indicate linewidths with resolution below 10 Hz. Can the authors describe how this is achieved?

The authors claim that the camera frame rate was the dominant contribution to the calibration signal sensitivity. It would be useful to see this analysis, as well as a more detailed sensitivity estimate that takes into account all of the parameters involved in the experiment (“pixel size, quantum efficiency, readout characteristics and so forth”).

The authors mention using a camera frame rate of 6000 fps for NV ESR characterization, but it is not clear what frame rate was used to collect the CASR data in Figs. 3-4. It would be helpful to clarify this.

What physical information can be gained from the phase of the observed NMR signal?

Version 1:

Reviewer comments:

Reviewer #1

(Remarks to the Author)

I am satisfied with the response and revisions. I recommend publication as is.

Reviewer #3

(Remarks to the Author)

The authors have addressed all of my questions, and I would support publication of the manuscript in its current form.

Reply to the reviewers of NCOMMS-24-26519-T

Reviewer #1 (Remarks to the Author):

The paper demonstrates imaging of NMR signals in microfluidic structures with a 10 μm spatial resolution, using widefield optical readout of a layer of NV centres in a diamond chip. Imaging of NMR signals using NV-diamond was previously reported but these previous demonstrations (Refs 9,26, 27) were limited in spectral resolution. Here the authors implement a protocol providing a dramatically improved resolution that in principle enables chemical identification. As such, it can be argued this paper reports the first genuine NMR imaging demonstration with NV-diamond. The amount of engineering that went into making these experiments possible is truly impressive, and extremely well documented in the supplementary material, which will make the paper a very useful contribution to the NV community. Overall, I think the paper a significant step in NV sensing technology. I recommend publication in Nat Comms after the authors have addressed my comments below. Many of them relate to details that are given in the supplementary material but in my opinion should be discussed in the main text.

1. In the introduction there should be an explicit acknowledgement of the previous NMR imaging demonstrations (Refs 9,26, 27) and a discussion of the spectral resolution and the ability (or inability) to detect chemical shifts in these previous works, and how the present work solves this limitation. This will help the reader to better appreciate the progress reported.

We agree with the reviewer and added a short discussion in the introduction (page 1) as well as a new chapter to the supporting information (Supplementary Note 1).

Main Text: *This is in strong contrast to previous NV-based work, where either static magnetic fields^{14,15} or nanoscale NMR signals^{16,17} were imaged, where time-resolved measurements were not needed. Importantly, nanoscale NV-NMR experiments are limited by broad resonance lines which is not a limitation in our present work³ (Supplementary Note 1).*

Supplementary Note 1:

Statistical vs Boltzmann polarized NMR

NV based NMR techniques may be separated into two categories based on the nature of the liquid state NMR signal they detect.

1. **Statistical NMR signals:** *A finite (open, sensing) volume containing thermalized sample spins, will spontaneously emit short-lived NMR signals. These signals arise from random fluctuations in the average spin orientation and have a lifetime limited by the average diffusion time of the sample spins through the volume. As the sample spins present in the detection volume exchange with the surrounding sample, the cumulative spin polarization decays and may randomly reappear with little or no correlation to the previous signal. The line widths of NMR spectra of this origin are thus limited to several kilohertz, depending on the sample diffusion coefficient and NV implantation depth (See Supplementary Figure 1). We approximate the expected linewidth according to Eq. 1 where τ is the average diffusion time through the detection volume as a function of the*

sample's diffusion coefficient (D) and the NV implantation depth.

$$\Gamma = \frac{1}{\pi \cdot \tau(D, NV\text{-depth})} \quad (1)$$

In general, experiments designed to detect this statistical NMR signal use ensembles of shallow NV centers on the order of a few of nanometers. This provides for extremely high magnetic spatial resolution and localizes the detection. Furthermore, due to the small dominant sensing volume resulting from the shallow NV layer ($r \sim 2 \times \text{avg. depth}$), the statistical polarization is very high and dominates of potential thermal NMR signals (Supplementary Figure 2), making it the ideal regime to apply this technique. Signals of this origin have been detected with pulse sequences designed for variance detection, using both photodiodes and cameras as detectors¹⁻³. Such measurements have proven invaluable in the study of surface dynamics and small volume samples due to their ease of use and minimal timing requirements, allowing for the use of slow detectors.

2. **Boltzmann NMR signal:** An external magnetic bias field, induces a preferential alignment of the sample spins along the B-field according to a Boltzmann distribution as a consequence of the Zeeman splitting of the spin energy levels. This Boltzmann polarization of the sample can be excited by applying a resonant radio frequency pulse to the sample, inducing a coherent precession around the B-field. The exchange of sample spins thus excited with spins surrounding the dominant detection volume does not affect the coherence time as long as the B-field is sufficiently homogeneous over the diffusion length scale of an experiment or over a sample containing structure. Therefore, the line width of a Boltzmann NMR signal is not limited by diffusion through the dominant detection volume. Instead, the homogeneity of the bias field becomes an extremely important factor in the design of the experiment and is the main limiting factor for our measurements.

For higher proton counts, the Boltzmann NMR signal becomes dominant over the statistical signal, which is why diamonds with NV layers on the order of μm and thus larger detection volumes are typically used for such experiments.

Supplementary Figure 1: Estimated diffusion limited line widths of statistical NMR signals. Left: Estimated diffusion limited line width of a statistical NMR signal based on the average diffusion time through the dominant sensing volume as a function of implantation depth. Diffusion coefficient of water at 25 °C. Right: Color-coded estimated diffusion limited line width as a function of diffusion coefficient and NV implantation depth

Supplementary Figure 2: Statistical and Boltzmann polarisation as a function of the number of spins in the detection volume. Boltzmann values are computed according to our experimental parameters (\sim room temperature, \sim 85 mT)

2. I think the conclusion is missing a quantitative discussion of the achieved spectral resolution, what it can enable beyond the water case (illustrative examples of chemical shifts that could be detected would be helpful), and what further improvements may be expected in the future.

We thank the reviewer for the remarks and added the requested discussion to the main text's discussion (page 9).

In contrast to previous nanoscale NV-NMR imaging work^{16,17}, our spectral resolution of \sim 3 ppm is only limited by the sample's coherence time¹. We are currently limited by magnetic field homogeneity and stability, which can be greatly improved by further engineering (e.g., by using a superconducting magnet³⁵). Thus, we envision chemical imaging on a per-pixel basis for highly parallelized NMR analysis in the future.

3. Fig. 2c: comment on the spatial features observed in the left part, which are not expected from the loop antenna used. Are these measurement artefacts?

We thank the reviewer for their keen observation, which we have discussed internally during the data acquisition. We currently treat this as an experimental artifact and attribute it to laser power distribution, as these patterns move with the laser spot. Determining the exact underlying mechanism, such as heating or NV-ionization, is considered beyond the scope of this work. We have added a clarifying comment to the main text.

Main text page 3: *We observe an additional pattern which may originate from local laser induced temperature variations. However, as it disappears for the following sensing schemes (Fig. 2f), the origin was not investigated further.*

And page 4: *We observe a similar large scale spatial variation of the SNR as in the Rabi image.*

4. Fig 2e: add a time scale for the inset graph, and specify the frame rate and time resolution, since the high frame rate is a key technical difference compared to previous works.

We thank the reviewer for this suggestion and amended the figure as requested (page 4).

5. Fig 2f: the SNR is plotted, and so the reader can't tell whether the spatial variations in SNR (by a factor of up to 2 it seems) are due to variations in signal or noise or both. I think it is important to show (possibly in the supplement) or at least comment on the flatness of the measured signal. Indeed, for an external calibration signal one would expect a uniform signal across the field of view, so variations in the measured signal would inform on the accuracy of the measurement. Probably related to this point is the fact the authors later (Fig. 4) normalise the NMR amplitude by the calibration signal amplitude. The reason for that normalisation should be explained.

We are grateful to the reviewer for bringing up this important point, which helped us improve the data analysis of this measurement to better capture the correlation between signal and noise. This improved analysis increased the flatness of the SNR response of our measurement. Figure 2 has been updated accordingly. We added a chapter to the Supporting Information (Supplementary Note 6) pointing out the complex interplay of experimental factors with our camera sensor.

Supplementary Note 6

Sensor response

Supplementary Figure 10 visualizes the complex interplay between laser power distribution and camera sensor saturation for our CASR measurements of the calibration signal. The saturation of our camera sensor is in a suboptimal regime in our measurements (Supplementary Note 5). We observe that this sometimes leads the camera sensor to generate disproportionately high noise in regions of very low light exposure. Simultaneously, regions with higher camera saturation produce a disproportionately low signal and noise. This results in an overall flat Signal to Noise Ratio across the sensor.

Supplementary Figure 10: Calibration signal image components. a, effective CASR calibration signal response. b, FFT noise floor of CASR calibration signal. c, laser spot illuminating the measurement area.

Furthermore, we adjusted the main text to clarify the reason for the S0/Scalib reference.

Using the calibration signal amplitude as a reference for the ^1H NMR signal amplitude (S_0) mitigates laser and MW-drive inhomogeneities and allows us to reconstruct the channel path of the microfluidic chip (Fig. 4a & b)

6. Fig. 3d-f: Comment on why there is a residual NMR signal (small peak at the expected frequency). Is it due to bleeding of the signal from the channel due to the point-spread function?

While the optical resolution likely plays a minor role, the size and extent of the bleeding signal match very well our simulations of the NMR signals which take the dipolar interaction of the NV-center and the sample spins into account. Here, we observe that the extent and amplitude of the leaked signal is a function of the sample geometry relative to the NV-center's location. We added a short discussion on the spatial resolution in the main text on page 5 and have extended Supplementary Note 3 (see also next question).

7. The spatial resolution ($\sim 10 \mu\text{m}$) is a key achievement of the paper but there is no data in the main text to support the claim. I think the linecut from Extended Data Fig. 8 (the $10 \mu\text{m}$ case only) should be shown in Fig. 3 of the main text.

We are grateful to the reviewer for bringing up this important point, which was not discussed adequately in our previous version. We amended Figure 3 as requested. We also added a discussion on the spatial resolution in the main text on page 5 and adapted the chapter in the Supporting Information (Supplementary Note 3).

Fig. 3. Optical widefield magnetic resonance microscopy (OMRM) ... g, Channel cross-section of the average SNR, indicating the spatial resolution for our $\sim 10 \mu\text{m}$ NV-layer.

Main Text (page 5): The spatial resolution of $\sim 10 \mu\text{m}$ is currently dominated by the magnetic resolution, which is a function of the NV layer thickness ($\sim 10 \mu\text{m}$). This is due to the dipolar interaction of the NV and the sample spins¹⁸, which is in agreement with our simulations and additional experiments performed with a thicker NV layer (Supplementary Note 3). Optical aberrations imposed by imaging through our diamond substrate likely play a minor role³⁴ [Nishimura2024].

Supplementary Note 3 (adapted part)

The spatial resolution of our OMRM technique is ultimately limited by the optical diffraction limit. Imaging through the bulk of our diamond sample, which has a thickness of approximately $\sim 500 \mu\text{m}$ and a high refractive index, introduces optical aberrations that limit the achievable resolution. As demonstrated by Nishimura et al. (2024), the use of a 0.7NA objective to image a field of view of $\sim 200 \mu\text{m}$ through a diamond with a thickness of $\sim 500 \mu\text{m}$ imposes an upper limit on the optical resolution of $\sim 3 \mu\text{m}$, which is below our magnetic resolution⁹. The magnetic resolution in our experiment, which we define as the extent to which the signal leaks beyond the microfluidic channel boundary, is determined by the depth of the NV layer. The radius of the dominant sensing volume for a given sensing location is approximately twice the average NV depth¹⁰. The overall spatial resolution is determined by both the optical and magnetic resolutions. When the NV-layer thickness is below the optical limit, the optical resolution predominates, while for layers thicker than the optical limit, the magnetic resolution becomes the dominant factor. We performed OMRM with a $\sim 10 \mu\text{m}$ as well as a $\sim 40 \mu\text{m}$ thick NV ensemble. The expected dominant sensing volume radius/resolution, therefore, is $\sim 10 \mu\text{m} / 2 \cdot 2 \approx 10 \mu\text{m}$ and $\sim 40 \mu\text{m} / 2 \cdot 2 \approx 40 \mu\text{m}$. In Supplementary Figure 4 we observe NMR signal leakage beyond the microfluidic channel boundaries. The extent of this leakage is in good agreement with the expected magnetic resolution. The signal obtained with the $\sim 40 \mu\text{m}$ NV ensemble leaks approximately $\sim 40 \mu\text{m}$ beyond the microfluidic channel walls, while the signal leakage is limited to $\sim 10 \mu\text{m}$ for the associated NV ensemble (Supplementary Figure 4). In these experiments, the NV principal axis is orthogonal to the direction of the microfluidic channel.

8. The total acquisition time for the NMR images shown in Figs. 2-4 should be given in the main text.

We added the requested information to the manuscript on page 5.

9. Fig. 4: the horizontal linecuts for the simulations are flipped in sign; this is very confusing and makes it hard to compare with experiment. Also the choice of some of the vertical linecuts is odd, some are mainly through the excluded data.

We adjusted Figure 4 as requested.

10. Fig. 4: What is the origin of the noise in the simulations?

We agree with the reviewer that this point warrants an explicit explanation. The simulation can best be thought of as a Monte Carlo simulation, where proton positions are repeatedly redrawn at random from within the sample volume. This inherent randomness in proton placement leads to fluctuations in the interaction amplitudes, causing the resulting simulation to appear noisy. We have amended the last paragraph the simulation chapter of the Supplementary Note 8, to better reflect this.

Because the proton positions are randomly generated, the simulated interaction amplitudes are not homogeneous across the chip and are subject to fluctuations. This leads to noise in the final outcome of the simulation.

11. The fitting procedure (Eq. 1) as explained in the main text is a bit confusing. The authors write “we fit a Lorentzian function (Eq. 1) to each pixel NMR spectrum” but Eq. 1 is complex-

valued and not the usual real-valued Lorentzian and so it is not immediately clear how the parameters are extracted. If I understand correctly, Eq. 1 is fitted to the complex-valued FFT of the real-valued time-domain data. Maybe this could be clarified in the main text.

We thank the reviewer for their remarks and added clarifications to this important point both in the main text on page 6 as well as in the Supplementary Note 7. We explain, how classical NMR can often rely on the usual real-valued Lorentzian by phase correcting their complex valued spectra and retaining only the real part. In our case, phase correcting is prohibitively expensive which is why we rely on the full complex valued Lorentzian allowing us to capture phase information rather than phase correcting. We describe the derivation of the complex valued Lorentzian function from a simple time domain signal model in accordance with Keeler's "Understanding NMR Spectroscopy". Furthermore, we added Supplementary Figures 11 and 13 to visualize important details about the signal model.

Main Text: For more detailed analysis, we fit a Lorentzian function (Eq. 1) to each pixel complex valued NMR spectrum resulting from the FFT of the time domain signal.

Supplementary Note 7 (adjusted parts)

In most NMR applications the complex valued spectrum, obtained from Fourier transformation of the time domain signal, is phase corrected. By retaining the absorption mode only, a simplified Lorentzian fitting function, containing only the real part may be used. Because of the inherent difficulty of phase correcting ~ 300-thousand complex valued NMR spectra, we instead fit the full complex value Lorentzian obtained by Fourier transforming an exponentially decaying signal. See Supplementary Note 7 lower half for more details and derivations.

Supplementary Figure 11: Non-scaled and scaled Lorentzian. Left: Lorentzian without scaling. S_0 is directly proportional to the integral over the real part. Right: Scaled Lorentzian. S_F is equal to the peak height. The product of $S_F \cdot \Gamma$ recovers S_0 and thus the integral.

Supplementary Figure 13: Possible complex valued Lorentzian model functions. All four Lorentzian model functions derived above (Supplementary Note 7).

12. Fig. 4d: there is a gradient from left to right in the f_0 maps explained by non-uniformity of the applied B_0 field, but in the T-junction there is a less trivial variation between the vertical channel and the top reservoir. Please comment on the origin. If it is due to magnetic susceptibility variations as mentioned in the following paragraph on linewidth, I think the explanation should be moved to the discussion of the f_0 maps.

The frequency and line width variations result from a complex interplay between global B-field gradients (permanent magnet), local B-field gradients (resulting from the magnetization of equipment such as the objective), and susceptibility mismatches. Reliably distinguishing these effects is rather difficult and we would like to avoid any premature discussion at this stage. In the main manuscript for this discussion, we have lowered the tone.

Reviewer #2 (Remarks to the Author):

In their manuscript, Briegel et al. introduce diamond-based wide-field nuclear magnetic resonance microscopy that makes use of a high-speed camera. The manuscript is well written and polished.

My main concern with this manuscript is that while the data is nice, it reads more like a methods paper, and the main advance is the use of a camera that allows fast data extraction. The novelty is not very high, as many aspects have been demonstrated already. Wide-field NMR using diamond has been performed before (e.g. see Ziem et al, Quantitative nanoscale MRI with a wide field of view, Scientific Reports volume 9 (2019)), and the group has already published on microscale NMR in microfluidics devices (Allert et al., Lab on a Chip 2022). Introducing a fast camera is interesting, and will be of interest to specialists in the community, but many of the presented results have been demonstrated before.

We respectfully disagree with the reviewer on the novelty of our work, and we suspect that the reviewer may have misunderstood the key point of our work. For that reason we added a new chapter in the SI to clarify this point (Supplementary Note 1).

The work of Ziem et al. and DeVience et al. performed nanoscale widefield NV-NMR imaging which is clearly different from our approach. The key difference is, that the previous work detected spin noise (random spin fluctuations that can be observed when the number of sample spins is small). The disadvantage of detecting spin noise is that liquid samples typically cannot be probed, and high chemical resolution (a few ppm) is difficult to achieve. This can be overcome by detecting the thermal polarization, which allows us to detect NMR signals of water in ppm spectral resolution. It comes with two key innovations:

CASR readout scheme needs high speed camera. In contrast to the previously published spin noise detection where slow cameras can be used (very similar to the DC quantum diamond microscope), detecting thermal signals with high spectral resolution the coherently average synchronized readout (CASR) scheme is used where stroboscopic readout over time is required. This is easily achieved with a photodiode, but the implementation of this approach in a camera has so far been elusive due to the fast readout.

The importance and novelty just of this step are evidenced by two papers that appeared during our review process: <https://arxiv.org/abs/2406.15450> and <https://arxiv.org/abs/2407.07025>. These papers focus on developing the detection and imaging of MHz signals with high-speed cameras (Fig. 2f in our paper). However, the key advance of our paper is the application of this technology to high frequency-resolution imaging of NMR samples:

High spectral resolution NMR on a widefield optical microscope. By using thermal signal detection and high speed camera readout, we achieve high spectral resolution NMR spectra on a camera for the first time. Although this may sound trivial after the implementation of the high-speed camera readout, it is a complicated technical endeavor (also appreciated by reviewer 1). To achieve high spectral resolution, the magnetic field homogeneity must be very high (ppm). Therefore, not only have we developed a new highly homogeneous permanent magnet assembly that fits the microscopy set up, but the optics must also be non-magnetic. These technical advances were not necessary for the previous work of Ziem and Deviance because of the low spectral resolution.

Thus we argue, that our “paper reports the first genuine NMR imaging demonstration”, as appreciated by reviewer 1.

A few detailed comments, though overall the manuscript is very clearly laid out:

- I could not find any details on the two steps that precede the CASR pulse sequence (shown in Figure 3.a): Overhauser hyperpolarization and ^1H excitation pulses. I may have missed it but could not see anything in the supplementary materials. Please add details and calibrations for these.

We agree with the reviewer that the two steps of ^1H excitation pulses and the Overhauser DNP drive were insufficiently described. We have amended the Methods section as well as the Supplementary Note 11 (Supplementary Figure 22), which describes all steps preceding the CASR NMR measurement, by including the calibration of the ^1H excitation pulse duration and heuristic of choosing the Overhauser DNP parameters, which results from citation 32 (Bucher2020, Hyperpolarization-Enhanced NMR Spectroscopy with Femtomole Sensitivity Using Quantum Defects in Diamond.).

Methods (page 13): *To detect the ^1H NMR signal, a proton $\pi/2$ pulse (duration: $\sim 200 \mu\text{s}$, power: $\sim 70\%$ of NV-drive, frequency: $\sim 3.5625 \text{ MHz}$) is applied to start the precession of the nuclear spins. **After initial detection of the NMR signal, the proton pulse time is swept to calibrate its optimal duration (Supplementary Fig. 22).** To increase the signal amplitude and reduce the averaging time, the sample is hyperpolarized using an Overhauser DNP pulse (duration: $\sim 400 \text{ ms}$, frequency: $\sim 2.34 \text{ GHz}$), which transfers the thermal polarization of electron spins in TEMPOL to the nuclear spins³². **As the effectiveness of the Overhauser DNP drive improves with increased power and duration, these parameters are heuristically selected as a balanced compromise to maximize efficiency while avoiding overheating and potential damage to the microwave equipment.***

Supplementary Note 11:

Supplementary Figure 22: Proton Rabi. The duration of the radio frequency pulse, used to excite the NMR signal, is swept in duration and the resulting amplitude of the NMR signal tracked. The $\pi/2$ -pulse duration is extracted as the first peak in the signal amplitude.

- The discussion around the linewidth could be misleading, as the first sentence states this

'provides information about the proton relaxation time'. In the regime reported here the linewidth is dominated by other effects, and it would be good to amend the statement, and perhaps provide estimates on what would need to be achieved for the linewidth to actually reveal proton relaxation times.

We think the question is probably related to the reviewer's general comment, which may indicate a misunderstanding. In contrast to previous work (Ziem et al., DeVience et al.), where the linewidth is broadened due to diffusion or dipolar interaction, our linewidth is actually limited by the proton relaxation time. The only limitation is the homogeneity and stability of the magnetic field (T_2^*). We added a discussion on the spectral resolution in the discussion of the manuscript and an additional chapter in the Supporting Information (Supplementary Note 1).

Main text discussion (page 9): *In contrast to previous nanoscale NV-NMR imaging work^{16,17}, our spectral resolution of ~ 3 ppm is only limited by the sample's coherence time¹. We are currently limited by magnetic field homogeneity and stability, which can be greatly improved by further engineering (e.g., by using a superconducting magnet³⁵). Thus, we envision chemical imaging on a per-pixel basis for highly parallelized NMR analysis in the future.*

Supplementary Note 1:

Statistical vs Boltzmann polarized NMR

NV based NMR techniques may be separated into two categories based on the nature of the liquid state NMR signal they detect.

1. **Statistical NMR signals:** *A finite (open, sensing) volume containing thermalized sample spins, will spontaneously emit short-lived NMR signals. These signals arise from random fluctuations in the average spin orientation and have a lifetime limited by the average diffusion time of the sample spins through the volume. As the sample spins present in the detection volume exchange with the surrounding sample, the cumulative spin polarization decays and may randomly reappear with little or no correlation to the previous signal. The line widths of NMR spectra of this origin are thus limited to several kilohertz, depending on the sample diffusion coefficient and NV implantation depth (See Supplementary Figure 1). We approximate the expected linewidth according to Eq. 1 where τ is the average diffusion time through the detection volume as a function of the sample's diffusion coefficient (D) and the NV implantation depth.*

$$\Gamma = \frac{1}{\pi \cdot \tau(D, NV\text{-depth})} \quad (1)$$

In general, experiments designed to detect this statistical NMR signal use ensembles of shallow NV centers on the order of a few of nanometers. This provides for extremely high magnetic spatial resolution and localizes the detection. Furthermore, due to the small dominant sensing volume resulting from the shallow NV layer ($r \sim 2 \times \text{avg. depth}$), the statistical polarization is very high and dominates of potential thermal NMR signals (Supplementary Figure 2), making it the ideal regime to apply this technique. Signals of this origin have been detected with pulse sequences designed for variance detection, using both photodiodes and cameras as detectors¹⁻³. Such measurements have proven invaluable in the study of surface dynamics and small volume samples due to their ease of use and minimal timing requirements, allowing for the use of slow detectors.

2. **Boltzmann NMR signal:** An external magnetic bias field, induces a preferential alignment of the sample spins along the B-field according to a Boltzmann distribution as a consequence of the Zeeman splitting of the spin energy levels. This Boltzmann polarization of the sample can be excited by applying a resonant radio frequency pulse to the sample, inducing a coherent precession around the B-field. The exchange of sample spins thus excited with spins surrounding the dominant detection volume does not affect the coherence time as long as the B-field is sufficiently homogeneous over the diffusion length scale of an experiment or over a sample containing structure. Therefore, the line width of a Boltzmann NMR signal is not limited by diffusion through the dominant detection volume. Instead, the homogeneity of the bias field becomes an extremely important factor in the design of the experiment and is the main limiting factor for our measurements.

For higher proton counts, the Boltzmann NMR signal becomes dominant over the statistical signal, which is why diamonds with NV layers on the order of μm and thus larger detection volumes are typically used for such experiments.

Supplementary Figure 1: Estimated diffusion limited line widths of statistical NMR signals. Left: Estimated diffusion limited line width of a statistical NMR signal based on the average diffusion time through the dominant sensing volume as a function of implantation depth. Diffusion coefficient of water at 25 °C. Right: Color-coded estimated diffusion limited line width as a function of diffusion coefficient and NV implantation depth

Supplementary Figure 2: Statistical and Boltzmann polarisation as a function of the number of spins in the detection volume. Boltzmann values are computed according to our experimental parameters (\sim room temperature, \sim 85 mT)

- In Figure 4.b) in the T geometry the simulation differs from measurement, particularly on the right hand side. Why is this? We thank the reviewer for this remark. We believe this variation to be a consequence of parameter coupling in the fully automated fitting procedure over approx. 1 million spectra. To support this claim, we would invite the reviewer to compare Figure 4 to Supplementary Note 7, Supplementary Figure 12, which is the analog to Figure 4 however based on purely numerical analysis without fitting. In this Supplementary Figure, the observed discrepancy is no longer present.

Supplementary Figure 12: Numerically analysed version of Fig. 4. NMR Signal amplitude referenced with calibration signal amplitude.

Reviewer #3 (Remarks to the Author):

The authors demonstrate widefield optically detected nuclear magnetic resonance microscopy by using ensembles of NV centers in diamond read out by a high-speed streaming camera. They modify an established NMR sensing technique to be compatible with camera read out, and then use the platform to image proton NMR signals associated with water in microfluidic chambers with approximately 10 μm spatial resolution. They then match the measured NMR signal to a dipolar simulation and see reasonable agreement.

Previous work in the field has measured NMR signals from small volumes, interrogating either single NV centers or NV ensembles without spatial information. This technical work extends to a widefield camera readout with spatial information engineered with a microfluidic structure, which may be applicable for applications in condensed matter or biological physics. The manuscript is clear and well-written, and all of the claims are well-supported by the data. The results are a sensible step forward for widefield NMR sensing, and the authors extensively benchmark future improvements to sensitivity and spatial resolution, which will be useful to other researchers in the NV NMR field. I believe the paper merits publication as long as the authors address the following minor comments:

The authors claim that the NMR spatial resolution is limited solely by the NV layer depth, but aberrations from imaging through the diamond may also be a strong contribution. How thick are the diamonds used, and how much do aberrations from imaging through the diamond with the 0.8 NA objective affect the spatial resolution of the measurements?

We thank the reviewer for this question which was not discussed adequately before. While it is true that imaging through a diamond substrate of considerable thickness over a wide field of view adds considerable aberration and reduces the optical resolution, this effect is limited to a $\sim 3 \mu\text{m}$ for our 0.5 mm thick diamond [Nishimura2024]. Thus, given the specifics of our experiment, the substantial NV-layer thickness of $\sim 10 \mu\text{m}$ remains the limiting factor (See Supplementary Figure 4). Optical aberrations due to the diamond become more important however, as the limit of resolution is further pushed, and the NV-layer thickness decreased. We have added a discussion on the spatial resolution to the main text on page 5 and have added the requested value for the diamond thickness on page 3. Furthermore we have updated Supplementary Note 3.

Main text (page 5): The spatial resolution of $\sim 10 \mu\text{m}$ is currently dominated by the magnetic resolution, which is a function of the NV layer thickness ($\sim 10 \mu\text{m}$). This is due to the dipolar interaction of the NV and the sample spins¹⁸, which is in agreement with our simulations and additional experiments performed with a thicker NV layer (Supplementary Note 3, Supplementary Figure 4). Optical aberrations imposed by imaging through our diamond substrate likely play a minor role³⁴.

Supplementary Note 3 (Updated part)

The spatial resolution of our OMRM technique is ultimately limited by the optical diffraction limit. Imaging through the bulk of our diamond sample, which has a thickness of approximately $\sim 500 \mu\text{m}$ and a high refractive index, introduces optical aberrations that limit the achievable resolution. As demonstrated by Nishimura et al. (2024), the use of a 0.7NA objective to image a field of view of $\sim 200 \mu\text{m}$ through a diamond with a thickness of $\sim 500 \mu\text{m}$ imposes an upper limit on the optical resolution of $\sim 3 \mu\text{m}$, which is below our magnetic resolution⁹. The magnetic

resolution in our experiment, which we define as the extent to which the signal leaks beyond the microfluidic channel boundary, is determined by the depth of the NV layer. The radius of the dominant sensing volume for a given sensing location is approximately twice the average NV depth¹⁰. The overall spatial resolution is determined by both the optical and magnetic resolutions. When the NV-layer thickness is below the optical limit, the optical resolution predominates, while for layers thicker than the optical limit, the magnetic resolution becomes the dominant factor. We performed OMRM with a $\sim 10\ \mu\text{m}$ as well as a $\sim 40\ \mu\text{m}$ thick NV ensemble. The expected dominant sensing volume radius/resolution, therefore, is $\sim 10\ \mu\text{m} / 2 \cdot 2 \approx 10\ \mu\text{m}$ and $\sim 40\ \mu\text{m} / 2 \cdot 2 \approx 40\ \mu\text{m}$. In Supplementary Figure 4 we observe NMR signal leakage beyond the microfluidic channel boundaries. The extent of this leakage is in good agreement with the expected magnetic resolution. The signal obtained with the $\sim 40\ \mu\text{m}$ NV ensemble leaks approximately $\sim 40\ \mu\text{m}$ beyond the microfluidic channel walls, while the signal leakage is limited to $\sim 10\ \mu\text{m}$ for the associated NV ensemble (Supplementary Figure 4). In these experiments, the NV principal axis is orthogonal to the direction of the microfluidic channel.

The T_2^* of the detected protons was 30 ms, but the data in Fig. 4e indicate linewidths with resolution below 10 Hz. Can the authors describe how this is achieved?

We thank the reviewer for bringing to our attention the typo in the footnote to Supplementary Table 4, in which we specified the lifetime of our proton signal as ~ 30 ms instead of the correct value of ~ 60 ms. The lifetime is estimated by applying a matched filter to a single (5 min) NMR measurement partially averaged across the sensor (Supplementary Figure 14). By using only a single acquisition, we preclude the need for thermal drift correction, which inevitably alters the signal linewidth.

Given this value we calculate the expected linewidth according to Keeler (Understanding NMR Spectroscopy) as $1 / (\pi * \text{signal lifetime}) = 1 / (\pi * \sim 0.06\text{s}) = \sim 5$ Hz.

Averaging more NMR measurements to achieve single pixel spectra requires thermal magnetic field drift correction as discussed in Supplementary Note 7. Drift correction often negatively impacts the signal linewidth, which leads to the increased linewidth we observe in Fig. 4.

The authors claim that the camera frame rate was the dominant contribution to the calibration signal sensitivity. It would be useful to see this analysis, as well as a more detailed sensitivity estimate that takes into account all of the parameters involved in the experiment (“pixel size, quantum efficiency, readout characteristics and so forth”).

We thank the reviewer for this remark. The discussion about what drives sensitivity is indeed a very important but also a complicated one. In recognition of the complexity and importance of this discussion, we have substantially expanded Supplementary Note 5. We have chosen to frame the discussion within the context of an experimental setting, where many camera parameters are dictated by the specifics of the experiment. We hope that this approach provides a practical and applicable guide for selecting a camera, making the analysis directly relevant to experimentalists.

Supplementary Note 5 (added parts)

Camera considerations

Selecting an appropriate camera for a quantum sensing experiment is a complex undertaking, requiring careful consideration of various experimental parameters. In this section, we will outline a step-by-step approach to determine the right choice of camera for our OMRM experiment (See Supplementary Figure 9). First, the required magnetic spatial resolution must be determined as a function of the sample in question. The thickness of the NV layer must be chosen appropriately, as this is the primary determining factor of the magnetic spatial resolution. At this point, the NV doping parameters can be adjusted to better match the diamond sensor to the application, which will influence the brightness and the coherence times. Together, these parameters determine the amount of signal-bearing fluorescence from the diamond per volume and per readout cycle. It is important to note that the amount of light that can effectively be used for readout is not a function of laser power, but rather depends entirely on the characteristics of the diamond. Given the selected magnetic resolution and the desired optical resolution, the next step is to select the appropriate optics. The choice of magnification, light collection efficiency, and field of view determine the spread and distribution of the fluorescent signal across the pixels. Finally, the camera must be selected to best match the constraints imposed by the previous choice of diamond and optics. Importantly, the pixel size (pixel pitch) should be matched to the chosen magnification and resolution of the optics in the image plane. In order to avoid limiting the resolution, oversampling is generally preferred. The full well capacity of the sensor is largely determined by the pixel size. Depending on how well the diamond, optics, and camera sensor are matched, the quantum efficiency of the sensor should be considered. Since the quantum efficiency typically ranges from $\sim 30\%$ to $\sim 90\%$, it affects the sensitivity of a measurement by a factor of up to $\sim \sqrt{3}$. Finally, the duty cycle of the experiment deserves consideration. The upper limit of the duty cycle in the case of dynamical decoupling sequences is imposed by the coherence time of the NV centers, which is typically in the range of 10-100 microseconds. It is advantageous to minimize the dead time between measurements. For example, for an experiment with a duty cycle of $100\ \mu\text{s}$, going from a very slow camera with a frame rate of 10 fps to a very fast camera with a frame rate of 10 000 fps, would increase the sensitivity by a factor of $\sqrt{1000}$. Furthermore, the limited lifetime of many physical phenomena of interest, such as that of an NMR signal, requires that the camera to be able to take an adequate number of measurements for each observation or risk extreme averaging times.

In summary: It is recommended to select the diamond and optical components in accordance with the specific requirements of the experiment, and then match the camera to these boundary conditions, rather than adapting the experiment to the camera's parameters. The choice of diamond and optics imposes strict constraints on the camera specifications. Once a rough match of the camera to these constraints has been achieved, the major difference between cameras lies in their achievable frame rates. Optimizing the frame rate can result in substantial increases, potentially on the order of $\sqrt{1000}$. However, it is important to note that this matching process is extremely challenging, and we have not achieved a perfect match in our reported work. Despite our best efforts, some deviations impacting readout quality remain (Supplementary Note 6), underscoring the difficulty of fully aligning the camera parameters with the experimental requirements (e.g. the diamond properties).

Supplementary Figure 9: Camera selection process

Supplementary Table 3: EoSense1.1MCX12-CM parameters

FPS ^a	Sensor	Pixel size	FWC ^b	QE ^c	Resolution	Color
3674	LUX13HS	13.7x13.7 μm^2	20.000	$\leq 30\%$	1.1 MP	Mono

^a Frames per second at full resolution.
^b Full Well Capacity.
^c Quantum Efficiency.

The authors mention using a camera frame rate of 6000 fps for NV ESR characterization, but it is not clear what frame rate was used to collect the CASR data in Figs. 3-4. It would be helpful to clarify this.

We thank the reviewer for this suggestion and amended the manuscript accordingly on page 4. All measurements in the main text were recorded at 6000 fps, which provides a reasonable tradeoff between the field of view and measurement speed.

What physical information can be gained from the phase of the observed NMR signal?

The dipolar nature of the interaction and the local nature of the sensor determine a relative phase of the detected signal. In our work we visualized that the phase is determined by local of the NV-center in respect to sample geometry. This, in principle, allows the reconstruction of the geometry by recording the phases. In future, we envision further application like high precession mapping of the local strength of applied gradient pulses, molecular diffusion or 3D imaging.

REVIEWERS' COMMENTS

Reviewer #1 (Remarks to the Author):

I am satisfied with the response and revisions. I recommend publication as is.

Reviewer #3 (Remarks to the Author):

The authors have addressed all of my questions, and I would support publication of the manuscript in its current form.

We would like to thank the reviewers for their positive feedback on our revised manuscript. Their comments were crucial in helping us to improve our work and we greatly appreciate their time and effort.